# Reliability and validity of the Japanese version of the Paediatric Pain Profile for children with severe motor and intellectual disabilities

**Mayumi Okita**[1]*, **Kaori Nio**[1], **Mayumi Murabata**[1], **Hiroaki Murata**[2], **Shotaro Iwamoto**[3]

**1** Child Health Nursing, Course of Nursing Science, Graduate School of Medicine, Mie University, Tsu, Mie, Japan, **2** Department of Pediatrics, National Hospital Organization Mie Hospital, Tsu, Mie, Japan, **3** Total Care Center for Children, Graduate School of Medicine, Mie University, Tsu, Mie, Japan

* mokita@med.mie-u.ac.jp

**Data Availability Statement:** All relevant data are within the paper and its Supporting information files.

## Abstract

Children with severe motor and intellectual disabilities experience chronic pain but cannot communicate verbally. However, no Japanese tool currently exists for assessing pain in this population. This study aimed to develop and evaluate the reliability and validity of a Japanese version of the Paediatric Pain Profile, which is a behavioral rating scale to assess pain in children with severe neurological disabilities. The sample comprised 30 children with severe motor and intellectual disabilities at three hospitals in Japan. Three specialist nurses rated low and high pain video scenes of the children (twice at 1-week intervals) using the Face, Legs, Activity, Cry, Consolability behavioral scale and a translated Japanese version of the Paediatric Pain Profile. On the basis of their ratings, we calculated the internal consistency, test–retest reliability, and intra- and inter-observer reliabilities of the Paediatric Pain Profile. Additionally, we assessed concurrent validity using the Face, Legs, Activity, Cry, Consolability behavioral scale and construct validity using low versus high pain scenes. Both internal consistency (low pain: alpha = 0.735; high pain: alpha = 0.928) and test–retest reliability ($r$ = 0.846) of the Japanese version of the Paediatric Pain Profile were good. Intra-observer reliability was substantial ($r$ = 0.748), whereas inter-observer reliability was only moderate ($r$ = 0.529). However, the concurrent validity with Face, Legs, Activity, Cry, Consolability scores was good ($r$ = 0.629) and construct validity was confirmed ($p$ < 0.001). We confirmed the validity of the Japanese version of the Paediatric Pain Profile, but reliable pain assessment may require repeated ratings by the same person. To accurately assess pain in children with severe motor and intellectual disabilities, healthcare staff must be properly trained and become more skilled in using the Japanese version of the Paediatric Pain Profile.

## Introduction

The concept of "severe motor and intellectual disabilities" (SMID) is a novel concept that originated in Japan. The definition of a person with SMID is an individual who is bedridden or

**Funding:** This study was supported by JSPS
KAKENHI Grant Number JP19K11031 and Mie
Medical Research foundation. The funders had no
role in study design, data collection and analysis,
decision to publish, or preparation of the
manuscript.

**Competing interests:** The authors have declared
that no competing interests exist.

able to sit, crawl, or walk with support, and has a profound intellectual disability (IQ < 35;
Diagnostic and Statistical Manual, Fourth Edition) [1]. Individuals with SMID have motor
abilities comparable to levels IV–V of the Gross Motor Function Motor Classification System
(GMFCS) [2], and they only communicate nonverbally or using body language. The concept
of SMID is very similar to that of severe multiple disabilities [3]. The main causes of SMID are
hypoxic encephalopathy/postnatal distress (20.0% of cases); unknown prenatal causes (10.7%);
external causes, including brain injury (8.3%); and meningitis/encephalitis (7.0%) [4]. There
are approximately 43,000 SMID patients in Japan, and their number is increasing. Some of
these patients require advanced home medical care, such as mechanical ventilation.

Previous studies show that children with severe neurological or cognitive impairments
experience chronic pain [5–10] from gastrointestinal dysfunction, musculoskeletal complica-
tions, and therapeutic procedures, as well as from other conditions common to all children
(e.g., otitis media, dysmenorrhea, or appendicitis) [7, 10, 11]. These children experience diffi-
culty in pain management, and their pain has been poorly recognized and inadequately man-
aged [12]. Despite analgesic use, the frequency of pain episodes and level of distress are
invariant over time, which indicates the inadequacy of treatment [13].

There have been no studies of day-to-day pain in SMID children in Japan because these
children have previously lived only in dedicated facilities or at home. Recently, children with
SMID who require advanced medical care (e.g., tracheostomy management, ventilator man-
agement, or tube feeding) have started to attend school or to go on outings. A common tool is
needed to explore the experiences of children with SMID as their social interactions increase.
However, there are no assessment tools to evaluate day-to-day pain in this population. The
Paediatric Pain Profile (PPP) is a 20-item behavioral rating scale that assesses pain in children
with severe neurological disabilities. Each item is rated on a four-point scale (0–3) as occurring
"not at all" to "a great deal" in a particular given time period. Hunt et al. have assessed the
validity and reliability of the scale [14].

Thus, we were able to develop a Japanese version of the PPP to assess pain in children with
SMID, and to evaluate several types of reliability and validity of the scale.

## Methods

### Translation

We used the back-translation method. In the first step, we obtained permission from the devel-
oper (Dr. Anne Hunt) and the copyright-holding facility (University College London Busi-
ness) to translate the PPP into Japanese. Next, two pediatricians, a palliative care doctor, and
two nursing researchers translated the PPP into Japanese. In the third step, we conducted a
pilot test on two children with SMID. In the fourth step, a professional bilingual (English and
Japanese) translator back-translated the Japanese version of the PPP into English. Only a few
minor changes were needed to the back-translated Japanese version of the scale. Therefore,
there were no substantial problems with the translation and a final Japanese version of the PPP
was produced (Fig 1).

### Procedures

First, 30 children were filmed during their daily nursing care. The care included one of nine
activities: suction, rehabilitation, dietary and oral care, repositioning, transfer, tracheal cannula
care, blood sampling, bed bathing, changing clothes. We recorded approximately 10–50 min-
utes of movies per child and extracted and edited low pain and high pain recordings from the
movie. Finally, each child's movie was edited to produce approximately 30 seconds of low pain
observations and approximately 30 seconds of high pain observations. One-minute movie

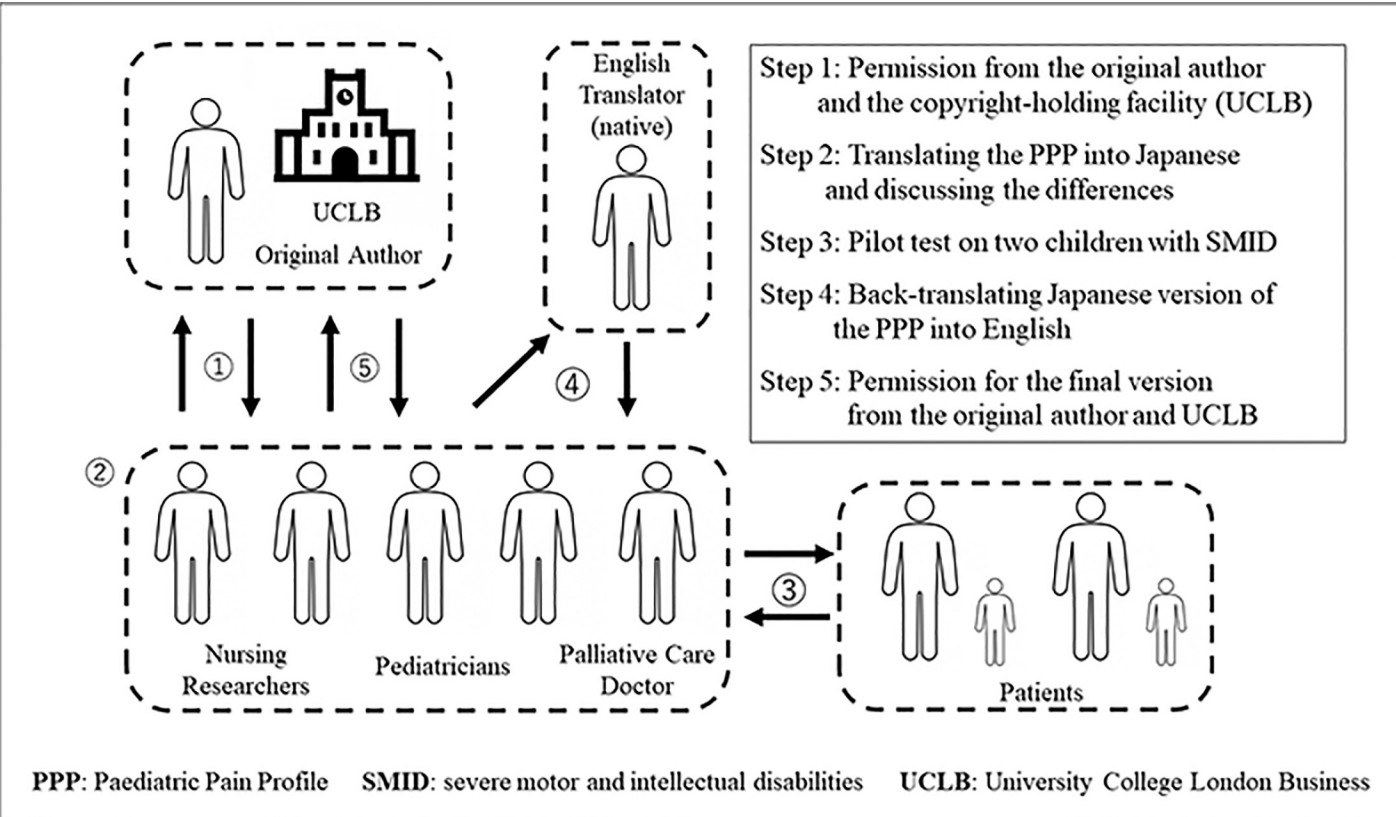

**Fig 1. Back-translation method used to translate the English version of the PPP into Japanese.** Abbreviations: PPP, Paediatric Pain Profile; SMID, severe motor and intellectual disabilities; UCLB, University College London Business.

episodes (i.e., 30 seconds of low pain and 30 seconds of high pain observations per child) were then individually rated by each of three evaluators using a 20-item Japanese version of the PPP.

Next, we recorded the following participant characteristics: age, sex, and current medical care.

Each of three specialist SMID nurses evaluated each child's pain using the Japanese version of the PPP and the Japanese version of the Face, Legs, Activity, Cry, Consolability (FLACC) behavioral scale [15], while watching a video recording of the child's daily care session. The FLACC behavioral scale has shown high reliability and validity for assessing acute pain in non-verbal children [16–18]. The Japanese version of the scale has also shown good to excellent reliability and validity. This scale comprises five items, each of which is rated on a three-point scale (0–2).

Finally, 1 week after the test, the same nurses used the PPP to evaluate each child's pain again while watching the same video recordings.

## Statistical analyses

For the analyses, we referred to methods for testing reliability and validity used in previous studies on the PPP [14, 19]. The rating of the video observations is a less common type of reliability check, so we could briefly mention the successful use of video observations by Hunt et al. (2004) (reference 14) to support our own use of this method.

The reliability of a scale is the extent to which measurements using that scale are reproducible; that is, whether measurements of individuals on different occasions, by different observers, or using similar or parallel tests produce the same or similar results [20]. To assess reliability, intra- and inter-observer reliability were examined using intraclass correlation coefficients (ICCs). Test–retest reliability was calculated using Spearman's rank correlation coefficient for the retest interval (1 week). Cronbach's alpha, a statistic that describes the average correlation between each of the items on a scale, was used to assess the internal consistency of the scale. Cronbach's alpha values of 0.7 or greater were considered indicative of excellent internal consistency. The kappa coefficient index of Landis & Koch [21] has often been used to assess ICC values. According to their guidelines, the strength of agreement indicated by an alpha of 0.00–0.20 is "slight"; 0.21–0.40 is "fair"; 0.41–0.60 is "moderate"; 0.61–0.80 is "substantial"; and 0.81–1.00 is "almost perfect".

To show that a test measures what it was intended to measure requires some evidence of validity [20]. Construct validity is the most important test and is based on logical positivistic assumptions, which require a coherent and well-articulated theory from which to ground validity claims [22]. We calculated the concurrent validity between the PPP and FLACC scores using Spearman's rank correlation coefficient. In addition, the construct validity of the PPP was evaluated using the Wilcoxon signed-rank test for paired data to compare scores between the "low pain" and "high pain" scenes. Comparisons at $p < 0.05$ were considered statistically significant.

All analyses were conducted using the Statistical Package for the Social Sciences (SPSS Inc., IBM Company, Chicago, IL, USA) version 25.0.

## Participants and sample size

Participants were children with SMID (1–18 years of age) who were patients at three specialized hospitals in Japan from February 2019 to August 2019. Exclusion criteria were the loss of one or more movements, of any facial expressions, or of any eye-catching expressions.

The sample size was based on a power calculation for proportions and was calculated based on the results of a previous study [14]. The correlation for the level of agreement between observers needed to be at least 0.6. We determined that a sample of 30 children would be required for $\alpha = 0.05$ and a test power $(1 - \beta)$ of 0.95.

## Ethics

The study was first approved by the medical ethics review board of Mie University (U2019-003) and subsequently approved by the medical ethics review board of each of the research hospitals (National Hospital Organization Mie Hospital; 30–35, MEIWA Hospital; G18, Mie prefectural Medical Center for Childgrowth Development and Disability; none). The parents of all subjects provided written informed consent before their participation.

## Results

### Characteristics

Of the total sample of 30 children, 23 had medical complexity. Table 1 shows the characteristics of children with SMID in this study.

### Internal consistency

Internal consistency was good, as shown by a Cronbach's alpha of 0.928 for "high pain" scenes and 0.735 for "low pain" scenes.

**Table 1. Characteristics of the children with SMID in this study.**

| Characteristics | n = 30 |
|---|---|
| Age (years) (mean ± SD) | 6.5 ± 4.3 |
| Sex (M/F) | 21:09 |
| **Medical Care** | |
| Ventilator | 11 (36.7%) |
| Tracheotomy | 13 (43.3%) |
| Nasal airway | 2 (6.7%) |
| Suction | 18 (60.0%) |
| Oxygen inhalation | 5 (16.7%) |
| Tube feeding | 23 (76.7%) |
| Frequent of position changes: more than six times per day | 20 (66.7%) |

Abbreviations: F, female; M, male; SD, standard deviation; SMID, severe mental and intellectual disabilities.

### Intra-observer reliability

Each of the three nurses watched and scored the pain-related recordings of the 30 children twice. Table 2 shows the analysis results for the ICC based on a 95% confidence interval for the agreement rate of the two evaluations using a one-way variance model.

The intra-observer agreement rate was $r = 0.748$. Using the criteria provided by Landis & Koch [21], which have often been used in previous studies, this agreement was judged to be substantial.

### Inter-observer reliability

Each of the three nurses watched and scored the pain-related recordings (of both the "low pain" and "high pain" scenes) of the 30 children twice. Table 2 shows the analysis results for the inter-observer agreement rates using a two-way mixed model, including the 95% confidence intervals and the ICCs. The inter-observer agreement rate was $r = 0.529$, which indicates moderate reliability.

### Test–retest reliability

Three nurses scored the 20 items of the PPP twice. The Spearman's rank correlation between the test–retest scores was $r = 0.846$, which indicates a very strong positive correlation (Fig 2).

### Concurrent validity

Three nurses evaluated the pain of 30 children using the PPP and FLACC scales. The Spearman's rank correlation between the PPP and FLACC scores was $r = 0.629$, which indicates a positive correlation (Fig 3).

### Construct validity

Table 3 shows the results of the PPP construct validity analysis, which was conducted using the Wilcoxon signed-rank test for paired data to compare scores between "low pain" and "high pain" scenes.

"High pain" scores were significantly higher than "low pain" scores ($p < 0.001$).

**Table 2. Intra- and inter-observer reliability of the PPP score.**

| Variables | ICC(1,2) [95%CI] | ICC(2,1) [95%CI] |
|---|---|---|
| 1. Is cheerful | 0.666 [0.533–0.766] | 0.364 [0.130–0.599] |
| 2. Is sociable or responsive | 0.504 [0.334–0.643] | 0.004 [-0.181–0.257] |
| 3. Appears withdrawn or depressed | 0.148 [-0.059–0.343] | 0.004 [-0.187–0.249] |
| 4. Cries /moans/groans / screams or whimpers | 0.693 [0.569–0.787] | 0.524 [0.302–0.719] |
| 5. Is hard to console or comfort | 0.563 [0.404–0.689] | 0.310 [0.078–0.555] |
| 6. Self-harms e.g. biting self or banging head | 0.009 [-0.214–0.197] | 0.000 [-0.184–0.253] |
| 7. Is reluctant to eat / difficult to feed | 0.621 [0.477–0.733] | 0.457 [0.222–0.671] |
| 8. Has disturbed sleep | 0.053 [-0.255–0.155] | 0.064 [-0.230–0.179] |
| 9. Grimaces / screws up face / screws up eyes | 0.518 [0.350–0.654] | 0.419 [0.186–0.642] |
| 10. Frowns / has furrowed brow / looks worried | 0.554 [0.394–0.682] | 0.214 [-0.010–0.470] |
| 11. Looks frightened (with eyes wide open) | 0.487 [0.302–0.623] | 0.464 [0.243–0.669] |
| 12. Grinds teeth or makes mouthing movements | 0.554 [0.394–0.682] | 0.331 [0.106–0.564] |
| 13. Is restless / agitated or distressed | 0.651 [0.515–0.756] | 0.459 [0.237–0.665] |
| 14. Tenses / stiffens or spasms | 0.663 [0.529–0.764] | 0.293 [0.069–0.531] |
| 15. Flexes inwards or draws legs up towards chest | 0.626 [0.483–0.737] | 0.217 [-0.001–0.464] |
| 16. Tends to touch or rub particular areas | 0.678 [0.549–0.775] | 0.044 [-0.145–0.292] |
| 17. Resists being moved | 0.639 [0.500–0.747] | 0.104 [-0.097–0.354] |
| 18. Pulls away or flinches when touched | 0.471 [0.294–0.617] | 0.062 [-0.130–0.311] |
| 19. Twists and turns / tosses head / writhes or arches back | 0.600 [0.450–0.717] | 0.487 [0.268–0.686] |
| 20. Has involuntary or stereotypical movements /is jumpy / startles or has seizures | 0.662 [0.528–0.763] | 0.393 [0.168–0.614] |
| Total PPP Score | 0.748 [0.641–0.826] | 0.529 [0.316–0.717] |

Abbreviations: ICC, intraclass correlation coefficient; CI, confidence interval; PPP, Paediatric Pain Profile.

## Discussion

In this study, a Japanese version of the PPP was developed and used by nurses to evaluate the pain of children with SMID in day-to-day care settings. The results demonstrated the reliability and validity of the Japanese version of the PPP.

The PPP reliability results were satisfactory, as indicated by the substantial intra-observer agreement (ICC = 0.748), the moderate inter-observer agreement (ICC = 0.529), the very strong positive test–retest correlation ($r = 0.846$), and the internal consistency of the scale

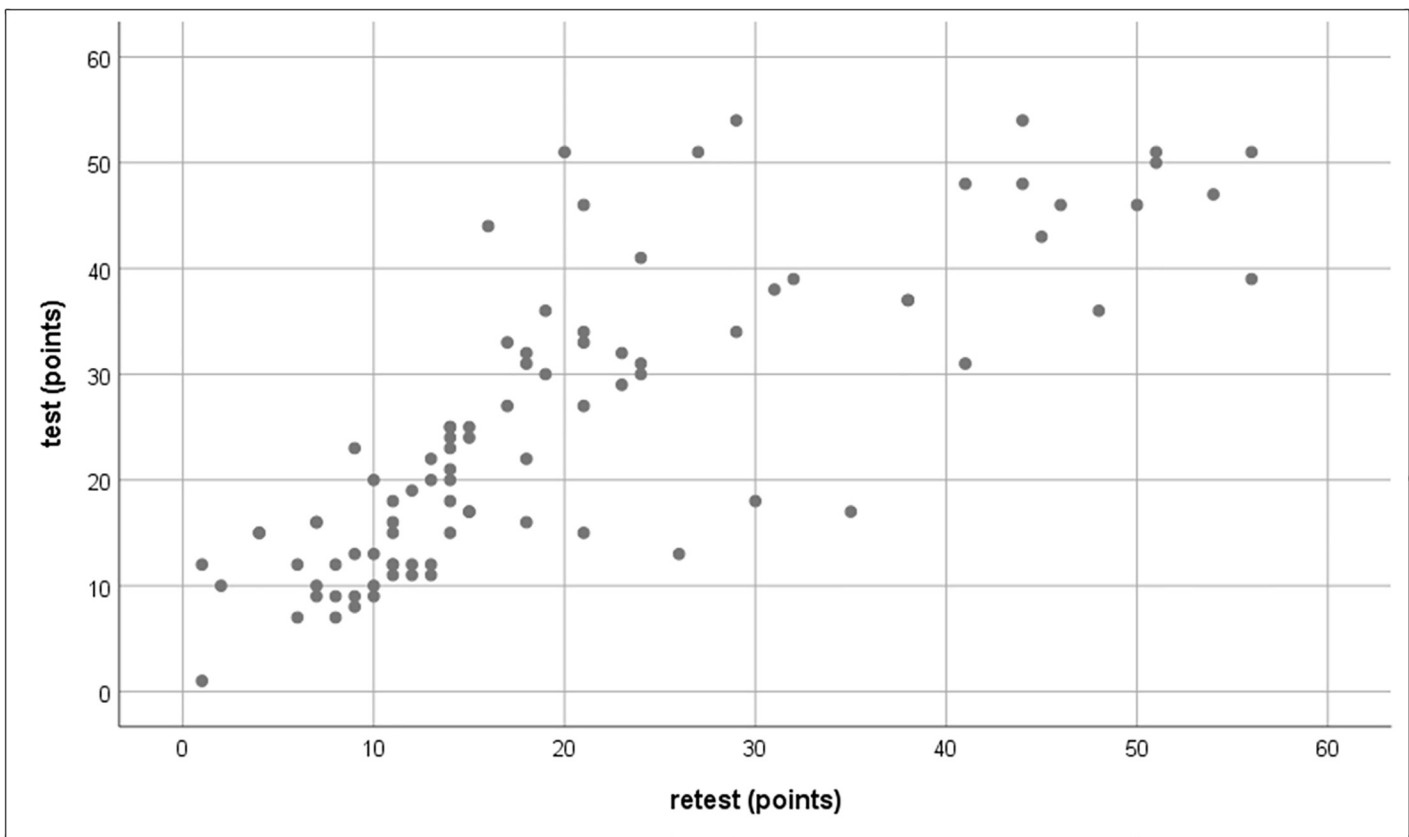

**Fig 2. Positive test–retest correlation for scores on the Japanese version of the PPP.** Abbreviations: PPP, Paediatric Pain Profile.

($a$ = 0.735–0.928). This suggests that if the same person (e.g., a parent, a physician, a visiting nurse, or a home care teacher) performed continuous evaluations, a more consistent evaluation would be obtained. The intra-observer reliability was better than the inter-observer reliability in this study. This was also the case in a previous study [19], in which the intra-observer reliability (ICC = 0.90) was higher than the inter-observer reliability (ICC = 0.62). In other words, PPP assessment is not suitable for cases in which the observer changes frequently, such as on a hospital ward, or for a single evaluation by more than one observer. The inter-observer reliability in this study was moderate because approximately 40% of the children were tracheostomized (dysphonic), which meant that it was difficult to evaluate PPP item 4 ("Cries /moans/groans/screams or whimpers"). In addition, approximately 75% of the children were tube-fed, so it may have been difficult to evaluate item 7 ("Is reluctant to eat/difficult to feed"). Items with an ICC < 0.4 were considered to contain terms that were confusing for the observers, or terms that did not correctly express the behavioral response to pain shown by children with SMID.

Regarding the validity of the scale, there was a strong positive correlation between the Japanese version of the PPP and the Japanese version of the FLACC, which assesses postoperative acute pain in non-communicative children. In addition, PPP scores increased significantly from "low pain" to "high pain" scenes. Therefore, we consider the Japanese version of the PPP to be a suitable tool for assessing pain in children with SMID.

Pain is always subjective. However, we must consider potential biases in self-reporting that reflect the level of cognitive function, expectations, and needs of the patient [23]. Nonverbal

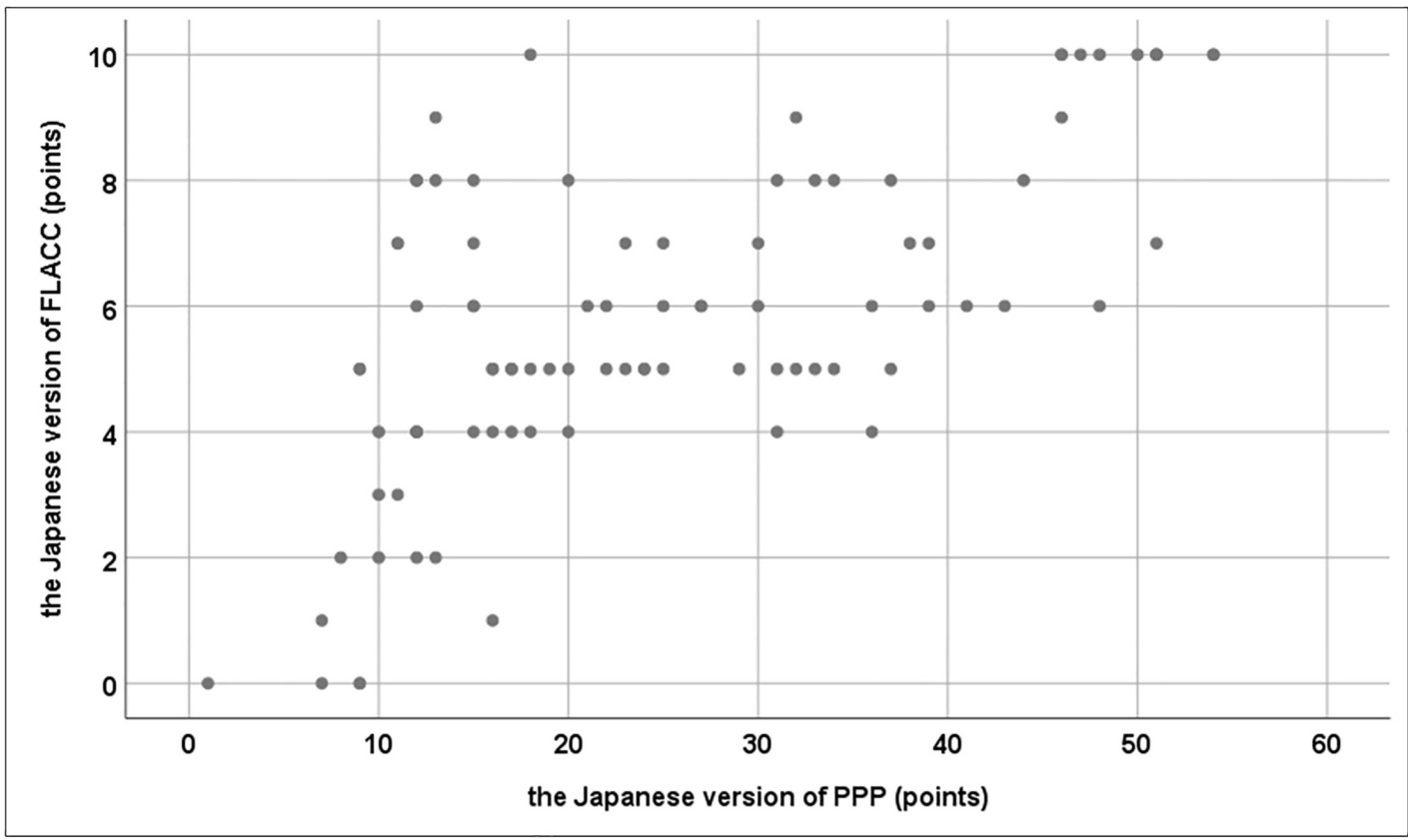

**Fig 3. Positive correlation between scores on the FLACC and the Japanese PPP.** Abbreviations: FLACC, Face, Legs, Activity, Cry, Consolability Behavioral Sscale; PPP, Paediatric Pain Profile.

expressions are generally considered more reliable than self-reports because they add context and meaning to the language and do not involve as much conscious control in language use [24]. Proxy assessments of pain in nonverbal children are also influenced by the children's ability to express themselves and by potential biases [25]. In addition, observers may differ in their assessments of pain based on their own particular experience, resulting in differences, for example, in how the child's behavioral response is perceived or an assumption that the child must be in pain.

Questions for future studies are why these nonverbal behavioral responses to pain appear; the sequence of events leading to pain-induced distress in the child; and how the observer can resolve the situation. To address these issues, we recommend that staff gain more experience in using pain scales, as Barney et al. [26] have suggested. Medical and support staff need to develop in parallel an educational program to operators and to parents. The present findings

**Table 3. Construct validity of the Japanese version of the PPP.**

|  | Low pain | | High pain | | |
|---|---|---|---|---|---|
|  | **mean** | **SD** | **mean** | **SD** | ***P*** |
| PPP score (points) | 2.89 | 3.56 | 25.21 | 13.91 | <0.001[a] |

Abbreviations: PPP, Paediatric Pain Profile; SD, standard deviation.

[a] Wilcoxon signed-rank test.

confirm that this scale can be applied to children with severe neurological or cognitive impairment, such as spinal muscular atrophy, and we hope that the Japanese PPP will be used to investigate pain in palliative care research.

## Conclusion

We demonstrated the reliability and validity of a Japanese version of the PPP. The results suggest that the Japanese version of the PPP is valid but that reliable assessment of pain in children with SMID may require repeated ratings by the same person.

## Supporting information

**S1 Fig. The Japanese version of the Paediatric Pain Profile.**
(TIF)

**S2 Fig. Paediatric Pain Profile.**
(TIF)

**S1 Table. This file contains all the data reported in the results.**
(XLSX)

## Acknowledgments

We thank Doran Amos, PhD, and Diane Williams, PhD, from Edanz Group (https://en-author-services.edanzgroup.com/ac) for editing a draft of this manuscript and helping to draft the abstract.

## Author Contributions

**Conceptualization:** Mayumi Okita.

**Data curation:** Mayumi Okita, Mayumi Murabata.

**Formal analysis:** Kaori Nio.

**Funding acquisition:** Mayumi Okita.

**Investigation:** Mayumi Okita, Mayumi Murabata, Hiroaki Murata.

**Methodology:** Mayumi Okita, Kaori Nio, Mayumi Murabata, Shotaro Iwamoto.

**Project administration:** Mayumi Okita, Kaori Nio.

**Resources:** Hiroaki Murata, Shotaro Iwamoto.

**Supervision:** Kaori Nio, Mayumi Murabata, Hiroaki Murata, Shotaro Iwamoto.

**Validation:** Mayumi Murabata.

**Visualization:** Mayumi Okita.

**Writing – original draft:** Mayumi Okita.

**Writing – review & editing:** Mayumi Okita.

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
