## [Decision Letter · Decision Letter 0]

7 Sep 2020

PONE-D-20-15345

Reliability and validity of the Japanese version of the Paediatric Pain Profile for children with severe motor and intellectual disabilities

PLOS ONE

Dear Dr. Okita,

Thank you for submitting your manuscript to PLOS ONE. After careful consideration, we feel that it has merit but does not fully meet PLOS ONE’s publication criteria as it currently stands. Therefore, we invite you to submit a revised version of the manuscript that addresses the points raised during the review process.

ACADEMIC EDITOR:

Please address the comments of the referees  and the additional comments provided below. 

1) Please ensure that the structure of the abstract follows the Plos One guidelines

2) Please ensure appropriate use of acronyms

3) Please ensure that methods are clearly explained (Eg use of FLACC scale as comparison. Is it a validated scale?may be good to make available the scale as appendix) 

We look forward to receiving your revised manuscript.

Kind regards,

Marzia Lazzerini, PhD

Academic Editor

PLOS ONE

Journal Requirements:

Additional Editor Comments (if provided):

1) Please ensure that the structure of the abstract follows the Plos One guidelines

2) Please ensure appropriate use of acronyms

3) Please ensure that methods are clearly explained (Eg use of FLACC scale as comparison. Is it a validated scale?may be good to make available the scale as appendix)

Reviewers' comments:

Reviewer's Responses to Questions

**Comments to the Author**

1. Is the manuscript technically sound, and do the data support the conclusions?

Reviewer #1: Yes

Reviewer #2: Partly

2. Has the statistical analysis been performed appropriately and rigorously? 

Reviewer #1: Yes

Reviewer #2: Yes

3. Have the authors made all data underlying the findings in their manuscript fully available?

Reviewer #1: Yes

Reviewer #2: Yes

4. Is the manuscript presented in an intelligible fashion and written in standard English?

Reviewer #1: Yes

Reviewer #2: No

5. Review Comments to the Author

Reviewer #1: I thank the authors for their contribution in implementing methods for assessing pain in children with SMID. Pain, as they affirm, is really the most important disruptive symptom in these patients, unfortunately frequently unsufficiently evaluated also for lack of suitable tools. Their proposal to develop a Japanese version of the PPP is really important. The work is well conducted in methods, their results are very interesting. As the authors conclude, it's important to underline that a good tool is not so good if the operators are not customized to use it or to evaluate the child. So, my suggest is to develop in parallel an educational program to operators and to parents as to take the best outcome from the local version of PPP.

Reviewer #2: 1. Please carefully edit the paper with a writing specialist for better quality language. Here are a few examples

(a) avoid using terms like the translation was a "success" in line 83 of the manuscript when describing the process of translation

(b) check with a methodologist on the validity of the phrase "highly significant" in in describing validity the abstract (line 40), (c) use the term "tool to measure pain" rather than "measure" in line 25 of the abstract. There are many more such inconsistencies. The term "measure" is usually a verb and is confusing when used as a noun.

(d) line 35: please use the term alpha instead of a or use the special symbol (greek notation) for alpha when reporting on Cronbach's alpha for reliability of the scale.

2. If SMID is a term that originated in Japan as the authors state in the introduction, there needs to be a citation. There also needs to be an explanation of how the SMID can be understood using the DSM or ICD diagnostic frameworks. How similar is the term SXI in the US educational system with SMID in Japan? There needs to be some contextual literature review to help the reader understand the use of the term SMID to cite it in future research.

3. Clarify procedure and ratings: Were all the children in the study video taped for low and high pain? So how many video clips were gathered for each child in the study? Did the evaluators rate all video clips or a random sample of the recordings coded as low and high pain?

4. Please provide clear operational definitions of forms of validity and reliability with citations that will be assessed in this study in the methods section. It will help the reader to then understand the results better. For example why did you choose to measure construct validity using a particular statistical test, cite from literature how this method is effective in evaluating the psychometric properties of a pain tool in a particular condition.

6. PLOS authors have the option to publish the peer review history of their article (what does this mean?). If published, this will include your full peer review and any attached files.

Reviewer #1: No

Reviewer #2: **Yes: **Preethy S. Samuel

---

## [Author Response · Author response to Decision Letter 0]

6 Oct 2020

October 3rd, 2020

Marzia Lazzerini

Academic Editor: PLOS ONE

PLOS ONE

Response to Reviewers [PONE-D-20-15345]

Dear Dr. Lazzerini,

Thank you very much for providing us with important feedback on our manuscript. We also appreciate the time and effort you and each of the reviewers have spent in providing insightful feedback on ways to strengthen our paper. We have made changes that reflect the detailed suggestions you have kindly provided. We hope that the revised manuscript and the responses provided below satisfactorily address all the issues and concerns you and the reviewers noted.

To facilitate your review of our revisions, the following is a point-by-point response to the questions and comments delivered in your letter dated 7th September 2020. 

Yours sincerely,

Mayumi Okita, RN, PHN, MSN

Child Health Nursing, Course of Nursing Science, Graduate School of Medicine, 

Mie University

Address: 2-174 Edobashi, Tsu, Mie 514-8507, Japan

Tel./Fax: +81-59-231-5282

E-mail: mokita@med.mie-u.ac.jp 

RESPONSE TO ADDITIONAL EDITOR:

Thank you very much for your review of manuscript # PONE-D-20-15345.

Comment 1: Please ensure that the structure of the abstract follows the Plos One guidelines

Thank you for your comment. We have confirmed that the structure of the abstract follows the PLOS ONE guidelines. We have also spelled out all the acronyms in the abstract.

Comment 2: Please ensure appropriate use of acronyms

Thank you for your comment. We have confirmed appropriate use of all acronyms in the manuscript. We have also spelled out all the acronyms in the abstract.

Comment 3: Please ensure that methods are clearly explained (Eg use of FLACC scale as comparison. Is it a validated scale?may be good to make available the scale as appendix)

Line 114: In response to the editor’s suggestion, we have changed the text “The FLACC behavioral scale is a well-known measure of acute pain that has been used by clinicians worldwide” to “The FLACC behavioral scale has shown high reliability and validity for assessing acute pain in nonverbal children [16–18].”

We have also added these papers to the References:

16. Merkel SI, Voepel-Lewis T, Shayevitz JR, Malviya S. The FLACC: a behavioral scale for scoring postoperative pain in young children. Pediatr Nurs. 1997 May-Jun;23(3):293–7. PMID: 9220806.

17. Malviya S, Voepel-Lewis T, Burke C, Merkel S, Tait AR. The revised FLACC observational pain tool: improved reliability and validity for pain assessment in children with cognitive impairment. Paediatr Anaesth. 2006 Mar;16(3):258-65. PMID: 16490089.

18. Kabes AM, Graves JK, Norris J. Further validation of the nonverbal pain scale in intensive care patients. Crit Care Nurse. 2009 Feb;29(1):59–66. PMID: 19182281.

RESPONSE TO REVIEWER 1:

Thank you very much for your review of manuscript # PONE-D-20-15345.

Comment 1: I thank the authors for their contribution in implementing methods for assessing pain in children with SMID. Pain, as they affirm, is really the most important disruptive symptom in these patients, unfortunately frequently unsufficiently evaluated also for lack of suitable tools. Their proposal to develop a Japanese version of the PPP is really important. The work is well conducted in methods, their results are very interesting. As the authors conclude, it's important to underline that a good tool is not so good if the operators are not customized to use it or to evaluate the child. So, my suggest is to develop in parallel an educational program to operators and to parents as to take the best outcome from the local version of PPP.

Thank you for your comments. We are delighted to hear that you think our work is important in the field. We agree with your suggestion to develop an educational program for operators and parents in parallel that includes guidance on how to use the PPP scale.

RESPONSE TO REVIEWER 2:

Thank you very much for your review of manuscript # PONE-D-20-15345.

Comment 1: Please carefully edit the paper with a writing specialist for better quality language. 

(a) avoid using terms like the translation was a "success" in line 83 of the manuscript when describing the process of translation

Line 93: In response to your suggestion, we have changed the text from “we were able to successfully create the Japanese version of the PPP” to “a final Japanese version of the PPP was produced.” 

(b) check with a methodologist on the validity of the phrase "highly significant" in in describing validity the abstract (line 40)

Line 43: We have changed the expression from “highly significant” to “confirmed.”

(c) use the term "tool to measure pain" rather than "measure" in line 25 of the abstract. There are many more such inconsistencies. The term "measure" is usually a verb and is confusing when used as a noun.

Line 28: Following your suggestion, we have changed the term from “measure” to “tool.”

Line 74: We have changed the text from “a common measure” to “a common tool.”

(d) line 35: please use the term alpha instead of a or use the special symbol (greek notation) for alpha when reporting on Cronbach's alpha for reliability of the scale.

Line 39: Thank you for this comment. We have changed the term from “α” to “alpha.”

Comment 2: If SMID is a term that originated in Japan as the authors state in the introduction, there needs to be a citation. There also needs to be an explanation of how the SMID can be understood using the DSM or ICD diagnostic frameworks. How similar is the term SXI in the US educational system with SMID in Japan? There needs to be some contextual literature review to help the reader understand the use of the term SMID to cite it in future research. 

Following your suggestion, we have explain the concept of SMID in the Introduction as follows:

Line 53: “The concept of “severe motor and intellectual disabilities” (SMID) is a novel concept that originated in Japan. The definition of a person with SMID is an individual who is bedridden or able to sit, crawl, or walk with support, and has a profound intellectual disability (IQ < 35; Diagnostic and Statistical Manual, Fourth Edition) [1]. Individuals with SMID have motor abilities comparable to levels IV–V of the Gross Motor Function Motor Classification System (GMFCS) [2], and they only communicate nonverbally or using body language. The concept of SMID is very similar to that of severe multiple disabilities [3]. The main causes of SMID are hypoxic encephalopathy/postnatal distress (20.0% of cases); unknown prenatal causes (10.7%); external causes, including brain injury (8.3%); and meningitis/encephalitis (7.0%) [4].” 

We have also added the following papers to the References:

1. Oshima K. Basic issue on severe motor and intellectual disabilities. Jpn J Public Health. 1971;35(11):648-55. (in Japanese)

2. Palisano R, Rosenbaum P, Walter S, Russell D, Wood E, Galupp B. Development and reliability of a system to classify gross motor function in children with cerebral palsy. Dev Med Child Neurol. 1997 Apr;39(4):214-23. PMID: 9183258.

3. Mensch SM, Echteld MA, Lemmens R, Oppewal A, Evenhuis HM, Rameckers EAA. The relationship between motor abilities and quality of life in children with severe multiple disabilities. J Intellect Disabil Res. 2019 Feb;63(2):100-12. PMID: 30175518.

4. Hanaoka T, Mita K, Hiramoto A, Suzuki Y, Maruyama S, Nakadate T, Kishi R, Okada K, Egusa Y. Survival prognosis of Japanese with severe motor and intellectual disabilities living in public and private institutions between 1961 and 2003. J Epidemiol. 2010;20(1):77-81. PMID: 19946176.

Comment 3: Clarify procedure and ratings: Were all the children in the study video taped for low and high pain? So how many video clips were gathered for each child in the study? Did the evaluators rate all video clips or a random sample of the recordings coded as low and high pain?

We have explained these details in the Procedures section, as follows:

Line 102: “First, 30 children were filmed during their daily nursing care. The care included one of nine activities: suction, rehabilitation, dietary and oral care, repositioning, transfer, tracheal cannula care, blood sampling, bed bathing, changing clothes. We recorded approximately 10–50 minutes of movies per child and extracted and edited low pain and high pain recordings from the movie. Finally, each child’s movie was edited to produce approximately 30 seconds of low pain observations and approximately 30 seconds of high pain observations. One-minute movie episodes (i.e., 30 seconds of low pain and 30 seconds of high pain observations per child) were then individually rated by each of three evaluators using a 20-item Japanese version of the PPP.”

Comment 4: Please provide clear operational definitions of forms of validity and reliability with citations that will be assessed in this study in the methods section. It will help the reader to then understand the results better. For example why did you choose to measure construct validity using a particular statistical test, cite from literature how this method is effective in evaluating the psychometric properties of a pain tool in a particular condition.

We have added the following text to the Statistical analyses section to clarify the reliability and validity:

Line 123: “For the analyses, we referred to methods for testing reliability and validity used in previous studies on the PPP [14,19]. The rating of the video observations is a less common type of reliability check, so we could briefly mention the successful use of video observations by Hunt et al (2004) (reference 14) to support our own use of this method.”

Line 127: “The reliability of a scale is the extent to which measurements using that scale are reproducible; that is, whether measurements of individuals on different occasions, by different observers, or using similar or parallel tests produce the same or similar results [20].”

Line 139: “To show that a test measures what it was intended to measure requires some evidence of validity [20]. Validity tests include content validity, criteria-related validity, and construct validity (among others), but it is usually sufficient to confirm two or more types of validity [22]. Construct validity is the most important test and is based on logical positivistic assumptions, which require a coherent and well-articulated theory from which to ground validity claims [23].” 

We have also added the following papers to the References: 

20. Streiner DL, Norman GR, Cairney J, editors. Health Measurement Scales. A practical guide to their development and use. 5th ed. New York: Oxford University Press; 2015.

22. Okaya K., Kawaguchi T. Scale or Measurement tool development process, and the procedure for creating the Japanese version (Japanese translation). Jpn J Nurs Sci. 1996;16(1):21-27. (in Japanese).

23. Kane MT. Validation. In: Brennan RL (Ed.), Educational Measurement. 4th ed. 2006;17-64.Westport, CT: American Council on Education/Praeger.

Again, we appreciate all your insightful comments and have tried hard to respond to each of them. Thank you for spending the time and energy to help us improve our manuscript.

---

## [Decision Letter · Decision Letter 1]

24 Nov 2020

Reliability and validity of the Japanese version of the Paediatric Pain Profile for children with severe motor and intellectual disabilities

PONE-D-20-15345R1

Dear Dr. Okita,

We’re pleased to inform you that your manuscript has been judged scientifically suitable for publication and will be formally accepted for publication once it meets all outstanding technical requirements.

Kind regards,

Marzia Lazzerini, PhD

Academic Editor

PLOS ONE

Additional Editor Comments (optional):

After careful revision I can say that you have duly answered all comments from the referee. I feel that the paper can be accepted for publication. However I do recommend to revise the following two aspects

1) Revise the following sentence in the methods section line 140 (added in last submission) "Validity tests include content validity, criteria-related validity, and construct 140validity(among others), but it is usually sufficient confirm or more types of validity[22]" I think this may not be methodologically correct (validy test depends also on the type of tool). I suggest to delete this sentence.

2) To further emphasise the message pointed out by the referee # 1 I would suggest to modify the last sentence of the abstract into something like "staff must be properly trained and become more skilled in using the Japanese version of the PPP " . The suggestion of referee # 1 could be added yo the discussion section ( develop in parallel an educational program to operators and to parents ). This is actually very important and would help correct translation of these findings into practice.

Reviewers' comments:

Reviewer's Responses to Questions

**Comments to the Author**

1. If the authors have adequately addressed your comments raised in a previous round of review and you feel that this manuscript is now acceptable for publication, you may indicate that here to bypass the “Comments to the Author” section, enter your conflict of interest statement in the “Confidential to Editor” section, and submit your "Accept" recommendation.

Reviewer #1: All comments have been addressed

Reviewer #2: All comments have been addressed

2. Is the manuscript technically sound, and do the data support the conclusions?

Reviewer #1: Yes

Reviewer #2: Yes

3. Has the statistical analysis been performed appropriately and rigorously? 

Reviewer #1: Yes

Reviewer #2: Yes

4. Have the authors made all data underlying the findings in their manuscript fully available?

Reviewer #1: Yes

Reviewer #2: Yes

5. Is the manuscript presented in an intelligible fashion and written in standard English?

Reviewer #1: Yes

Reviewer #2: Yes

6. Review Comments to the Author

Reviewer #1: (No Response)

Reviewer #2: The authors have been responsive to all comments and I wish them the bets in publishing this manusript.

7. PLOS authors have the option to publish the peer review history of their article (what does this mean?). If published, this will include your full peer review and any attached files.

Reviewer #1: No

Reviewer #2: No

---

## [Editor Report · Acceptance letter]

11 Dec 2020

PONE-D-20-15345R1 

Reliability and validity of the Japanese version of the Paediatric Pain Profile for children with severe motor and intellectual disabilities 

Dear Dr. Okita:

I'm pleased to inform you that your manuscript has been deemed suitable for publication in PLOS ONE. Congratulations! Your manuscript is now with our production department. 

Kind regards, 

on behalf of

Dr. Marzia Lazzerini 

Academic Editor

PLOS ONE